# Happiness at Work among Public Relations Practitioners in Spain

**DOI:** 10.3390/ijerph19073987

**Published:** 2022-03-27

**Authors:** José Antonio Muñiz-Velázquez, Alejandro José Tapia Frade, Javier Lozano Delmar, Puri Alcaide-Pulido, Andrés del Toro

**Affiliations:** Department of Communication and Education, Universidad Loyola Andalucía, 41704 Seville, Spain; jamuniz@uloyola.es (J.A.M.-V.); ajtapia@uloyola.es (A.J.T.F.); palcaide@uloyola.es (P.A.-P.); atoro@uloyola.es (A.d.T.)

**Keywords:** happiness, well-being at work, public relations, corporate communication, purpose, job satisfaction, communication consultants, engagement at work, organizational development

## Abstract

Happiness at work is a consolidated topic. Perhaps the PR and communication sector, often at the forefront of organizational change, is one of the industries where most progress has been made in this regard. The objective of the present study was to carry out an exploratory analysis on the extent to which PR is a profession that enables the development of happiness in the workplace. To this end, a questionnaire was administered to a sample of PR professionals in Spain (N = 256). The questionnaire consisted of the PERMA-profiler, a model where work relationships, engagement, positive affections/emotions, vital sense/purpose and achievements are measured. The results show a remarkable level of happiness among surveyed professionals, especially among women, who obtained higher scores on all five factors, although with a statistically significant difference only in two of the five factors in PERMA (Engagement and Relationships). Neither age nor the hierarchical level of the respondent had any incidence. Therefore, PR can be a profession that notably enables human flourishing at work, even more so among women.

## 1. Introduction

Happiness at work is something that, fortunately, is far from being a fleeting fad. Today, from the academic and research field, we find ample evidence that happiness, both hedonic and eudaimonic, and subjective, psychological and social well-being, are central axes for organizational development [1]. Likewise, from the business and professional perspective, there are also many industry sectors and small, medium and large companies that have placed the psychological well-being of their employees, and the rest of their stakeholders, at the center in terms of management. This is aligned with Díaz, Dearco and Arbeláez [2], when they claim that any feasible management of organizational happiness must reach all stakeholders, but it must always start with the employees. It is worth remembering that although happiness at work includes job satisfaction, it goes much further, as mentioned a few years ago by Fisher [3]. Now, in terms of the concept of organizational happiness or happiness at work, Moccia [4] speaks of a lack of consensus regarding its definition or configuration. However, rather than a lack of consensus, perhaps one could consider the coexistence of different models, given the multidisciplinary and complex nature of the phenomenon [5,6]. In any case, the different approaches seem to agree on the most essential aspects.

Thus, with emphasis on one or another aspect, overall happiness at work, or what can also be called human flourishing [7], consists of three main pillars: 

Firstly, subjective well-being, mainly around satisfaction with the work itself, as well as related positive emotions. 

Secondly, psychological well-being, concerning personal, intellectual and cognitive growth and development, autonomy, life purpose, among others. 

Thirdly, social well-being, grounded, among other things, on the positive and nourishing relationships that a job offers [8,9]. 

In short, according to Krekel, Ward and de Neve [10], a job is good if it is interesting, lacks excessive stress and high risks, involves good relationships, especially with bosses, and does not interfere excessively with family and personal life. One might say that it should go beyond a mere lack of interference, to allow for the possibility of mutual enrichment between work and personal life. Given that, as Weziak-Bialowolska et al. [11] emphasize, the overlap between well-being at work and well-being in life, in general, is indisputable. This is especially true if one considers the purpose of life and its day-to-day implementation, these authors state.

In short, although some emphasize subjective well-being [12], when talking about happiness at work, a series of aspects are brought into play that go much further, and which are related to psychological well-being or eudaimonic happiness. Although it has centuries of history, eudaimonia remains a multi-faceted and complex concept. Broadly, it would involve happiness based on universal values and virtues, on the development and cultivation of a vital sense, on growth as a person, human excellence, etc. [13]. However, eudaimonia and hedonia—or subjective well-being—are not only non-conflicting, but for the sake of the fulfillment of life as human beings, they should complement and nourish each other [14].

At this point, among the multiple existing models to approach the observation of happiness at work, and in life in general, the present study adopted a holistic approach where subjective well-being or hedonic happiness, psychological well-being or eudaimonic happiness, and also social well-being are incorporated. This is related to the concept of human flourishing, defined by Seligman [7] as that psychological growth and personal development that brings us closer to the fullness of life in psychological and psychosocial terms, and that is articulated, according to the same author, around different factors or key ingredients of our lives. These factors are summarized in a five-pillar model called PERMA [7]. Specifically, and in the employment domain, its five pillars could be summarized as follows:

P: Positive emotions: positive emotions that are felt and experienced throughout the day, in this case, in the work environment. It has to do with mental and emotional well-being, self-acceptance and perceived autonomy.

E: Engagement: the degree to which the person feels positively involved in the day-to-day tasks of his/her job, whether large or small, resulting in an overall commitment to his/her work. It is also related to the balance between professional and personal life and the environmental domain. Authors such as Kaul and Sen [15], among others, have emphasized the mutual relationship between engagement and happiness at work.

R: Relationships: quantity and quality of interpersonal relationships that are positive and nourishing for the well-being of the person at work. This refers to social well-being.

M: Meaning: significance or purpose that we attribute to our life. Thinking and feeling that our life in general responds to a purpose that transcends ourselves, and where our work plays a prominent role, given the time we devote to it in our lives. Under this model, we refer to the more spiritual dimension of human happiness.

A: Accomplishment: close to the concept of self-fulfillment. To feel and realize that a good part of the goals and objectives set are achieved, thus growing as an individual, and in our case, it includes the professional field.

In short, to achieve that human flourishing, within and thanks in part to our work, we would need: (1) to feel more positive emotions than negative ones during our working day; (2) to find ourselves with tasks that engage us per se, because we enjoy them; (3) to be surrounded in our work by people with whom we can build positive, nurturing, enriching relationships; (4) to have a relatively solid vital purpose, and to connect it, at least in part, with our profession, with our job and with the company for which we work; (5) and finally, to achieve the professional goals and objectives that we set for ourselves, and those that make us grow and achieve self-fulfillment in our career, and also as individuals.

With regard to the possible difference in work happiness between male and female professionals, also in PR, some differences could be expected in favor of women, if we take into account previous studies in other professional sectors, and with alternative theoretical models of approach [16]. As far as age is concerned, the variation in terms of happiness throughout life has been long studied [17]. Despite this, it is still a controversial issue [18], because there are many factors that may be mediating between happiness and lifespan [19]. Especially if it is limited to the work setting. In that sense, it is important to think about promotion on the hierarchical scale within the company, something roughly related to age. In any case, given the eudaimonic load of the PERMA model, one would expect a relatively greater degree of happiness in professionals who are older, more mature and experienced, whose work expectations are more in line with reality. In that sense, and related to age, it is worth asking whether the hierarchical level held in the agency could influence or be related positively in some way to happiness, following Garzon Castrillon et al. [12].

In short, whether or not gender, age, or hierarchical position in the agency are involved, this study aimed to provide evidence to demonstrate the extent to which we can argue that PR is a profession that facilitates or promotes a more or less integral happiness, that is, hedonic, eudaimonic and social, to people who choose to develop their professional career in this sector. The Spanish geographical area provides the first sample of this approach. In short, it was a matter of verifying whether it is true that the happiness a person hopes to find in their job when he/she decides to embark on a professional career in that field [20] surfaces later, once the person is already immersed in the profession.

This research is relevant and timely due to the globally competitive nature of the PR landscape [21]. This paper contributes to the limited literature focusing on the PERMA model in the PR sector and seeks to critically explore and understand the happiness or human flourishing of its professionals in the performance of their work. This research is exclusive in nature. As for recent studies on happiness at work by sector, we can find research looking at jobs in the education field [22], in the health and hospital sector [23], as well as small and medium sized commercial enterprises (SMEs) [24]. We can even find studies on happiness at work, that apply the PERMA model to a sample of professional musicians [25]. However, in the specific field of communication and PR, there are no studies focused entirely on happiness at work. The closest study found is the one conducted by Place [26], who focused on the moral development of PR professionals and its importance in the context of communication as an industry. Thus, the present work explores the well-being held by PR professionals in Spain regarding differences between gender, age, and held position because the working and social environment is a relevant source of satisfaction [3] or dissatisfaction at the workplace.

This means that the self-perceived happiness of the respondents, their subjective well-being or hedonic level, is slightly lower than their psychological well-being observed by the PERMA variables, that is, eudaimonic happiness. It is determinant for institutional development [1] and managerial implications of organizational happiness, especially for the employees [2,3]. In addition, the present work contributes to paying attention to the reality of the daily work of PR professionals. In terms of limitations, although the population studied is representative, the sample should be larger. Related to future studies and recommendations, we could complement the PERMA analysis with other scales that include psychological factors.

## 2. Hypothesis and Method

In line with previous findings, the general objective of this study is to determine whether PR professionals find a desirable level of happiness and human flourishing in their profession. At the same time, we wanted to verify the influence or interference that the aforementioned factors could have on it, such as sex, age, or the hierarchical level occupied within the agency or company. Thus, and always starting from the cited PERMA model of human flourishing [7], the following hypotheses were posed:

**Hypothesis 1** **(H1).***PR is a profession that enables human flourishing and happiness at work*.

**Hypothesis 2** **(H2).***Female PR professionals are happier at work than their male counterparts*.

**Hypothesis 3** **(H3).***The older the PR professional, the greater the happiness and human flourishing at work*.

**Hypothesis 4** **(H4).***The position or hierarchical level affects the happiness of PR professionals at work*.

According to the PERMA model, several tools or questionnaires have been developed, among which the scales by Kun et al. [27] stand out, as well as the PERMA-profiler by Butler and Kern [28]. For the present study, the latter was chosen, as it has been validated in English and Spanish [29,30]. Likewise, it has a version focused on the workplace, which has been adapted and validated by Watanabe et al. [31], the Workplace PERMA-profiler. 

Thus, the mentioned scale consists of 23 items with an 11-point scale, with choices between 0 and 10 (see Appendix A). Fifteen of the items are distributed equally among the five factors in the model, to which three other items are added to evaluate health (H), in order, among other things, to rule out possible problems that could be distorting the measurement. An additional item is added to measure the feeling of loneliness (L), and another one that asks about happiness of a more hedonic nature or subjective well-being in general, in our case related to work (Hap).

Finally, three more items are added to the scale that will measure negative emotions (NE), so that they can be contrasted with the rest of the items to check the validity of the answers for the rest of the questionnaire, while at the same time obtaining a measurement, albeit an inverse one, of subjective well-being or hedonia beyond the aforementioned item. Thus, as Goodman et al. [32] state, the PERMA measurement effectively concurs with subjective well-being (SWB). 

For the present study, therefore, a questionnaire was implemented consisting of the Workplace PERMA-profiler, together with other socio-demographic and employment issues, delivered online through Qualtrics. The sample of professionals who participated belonged to the database of Spanish companies or agencies within the Association of Consulting Companies in Public Relations and Communication in Spain (ADECEC for its initials in Spanish), and which comprised diverse professional profiles, covering the usual professional spectrum of the industry. All legal and ethical requirements were met, and the Ethics Committee of the University Loyola Andalucía supervised the process. The sample members that correctly completed the entire questionnaire totaled N = 256, and included mostly female professionals, with the percentage of women accounting for 68.3% of total respondents, compared to 31.7% male respondents. The average age was 36 years, ranging from 22 to 59 years. It should be noted that communication agencies in Spain, as a whole sector, employed around 2000 people at the time of the study (years 2019–2020), thus the degree of representativeness of the sample of this group is remarkable.

## 3. Results

After analyzing the results, the first aspect to note is that the whole sample had a mean of M = 7.26 (SD 1.26) in the Workplace PERMA-profiler scale. Table 1 shows the results in detail. If we consider that the maximum was 10, we could confidently accept H1. In other words, PR provides sufficient elements, always under the PERMA model, to notably develop the happiness or human flourishing of its professionals in the performance of their work. It is also true, however, that there is room for improvement to reach 10. On the other hand, it is interesting to note that the five factors of the PERMA model appeared well-balanced in the sample as a whole.

That is, in the items related to positive emotions, engagement, positive relationships, meaning and achievements, the scores obtained in the sample were even (see Table 1). It is true that the relationship factor stands out at the top, while meaning was the factor that remained below, although the difference between the five factors was not statistically significant.

Secondly, with respect to the hypothesis relating to gender difference, higher scores were found among women on all five parameters, with a statistically significant difference in two of the five factors in PERMA, engagement (E) (*p =* 0.029) and relationships (R) (*p =* 0.000). Statistically significant differences were also found in favor of women in the self-assessment of health (*p =* 0.040) (see Table 2). Therefore, one could partially accept H2, at least partially (see Table 1). That is, with respect to men, it may be partially suggested that women who work in PR find greater happiness in their work, fundamentally thanks to a greater commitment and increased enjoyment of relationships that they perceive as nourishing. However, it should be again noted that only in two of the five PERMA dimensions it was founded that the differences were statistically significant. In terms of age and its relationship to happiness in the work of PR professionals (H3), it can be said that no significant differences were found between different age and happiness groups. Therefore, H3 would be rejected, that is, in the level of happiness in general and in each of the five PERMA factors observed, no differences were found according to age. Finally, with respect to H4, it is also rejected, since no significant differences were found according to the position or hierarchical level of the participant. Therefore, in general terms, happiness in the work of PR professionals is independent of the rank or position held.

Beyond the five factors of the PERMA model, the scale measured other issues. Thus, the three items referring to health resulted in a mean of M = 7.02 (SD 1.74), with no significant differences in age or position. There were, however, differences between genders, with women displaying a higher level of health. Therefore, it can be said that the sample as a whole felt remarkably healthy, especially female respondents. 

As for the items measuring negative emotions associated with work and completing the measurement of the Workplace PERMA-profiler tool, the overall mean was M = 4.81 (SD 1.95), without differences of any kind between subgroups. Taking into account that we are dealing with inverse items, where the lowest score is desirable in principle, negative emotions at work were present in a very moderate way, below the midpoint. The item that measures the feeling of loneliness at work was even lower. In this case, an average of 3.67 was found, although the standard deviation was slightly higher than normal (SD 2.36). That is, since the feeling of loneliness at work is generally low, there is greater heterogeneity of cases in this sense. Finally, in the face of item 23, which asks directly about the subjective happiness or well-being considered to be at work (“Considering everything as a whole, how happy would you say you are at work?”), the total mean was 6.88 (SD 1.89), slightly below the PERMA measurements, but not in excess, nor significantly so. This means that the self-perceived happiness of the respondents, their subjective well-being or hedonic level, is slightly lower than their psychological well-being observed by the PERMA variables, that is, eudaimonic happiness.

## 4. Discussion

Once the results have been outlined, we can conclude that public relations professionals in Spain, specifically from a representative sample drawn from the entire population of people working in communication agencies in Spain, are certainly capable of finding in their work a rich source of happiness and human flourishing. It is especially so because, in general terms, they always feel loved, appreciated and valued by all their colleagues, receiving support from them and ultimately showing great satisfaction with their relationships in the office. The four other pillars of the observed human flourishing model also help them feel happy [7]. That is, there are more positive than negative emotions in their day-to-day work, they like their job and enjoy most of the tasks they perform, they are able to find their vital purpose, or at least part of it, in their work, and they feel they have enough margin to achieve career goals that allow them to grow as professionals and as people.

With regard to the differences between men and women, and in accordance with previous studies [16], we should emphasize that in the measurement as a whole, women always scored higher. We also found statistically significant differences in two of the five key factors of the PERMA model. There is also a statistically significant difference in favor of women in their self-perception of health. Thus, female public relations professionals feel significantly more absorbed in the work they are doing, i.e., more involved in the tasks carried out in general. They are also more cheerful and feel in better health, but above all more accompanied, supported and loved, and in general more satisfied with their relationships at work than their male colleagues.

As for the age difference and its relationship to happiness at work, no difference was found. The variation of happiness throughout life has been studied for a long time [17]. Despite this, it is still a controversial issue [18], because there are many factors that may be mediating between happiness and lifespan [19]. Even more so if we restrict it to the work environment. In that sense, let’s think about the promotion of individuals in the hierarchical scale within the company, something roughly related to age. It is not by chance, therefore, that no significant differences in happiness were found according to the hierarchical level. In any case, beyond the gender differences detected, the dataset portrays remarkably happy professionals in their work. That is to say, satisfied with a profession and/or job where they find sufficient well-being that constitutes labor well-being, both subjective and psychological, and always under the PERMA model by Seligman [7]. It seems, however, and as already noted, that there is room for improvement in all dimensions. That is, if they were school grades, it would indicate a low “B”.

Nevertheless, it is convenient to point out that the data extracted from the sample are similar to other studies that have used a similar measurement system, although at a general vital level, and that, limiting the observation to the work environment, the results are clearly superior to other studies in the same line. Looking at some of the other studies carried out in other sectors and latitudes, the first thing to note is that, with this particular tool, the PERMA-profiler, both in its version of general happiness and in its adaptation to the workplace (Workplace PERMA-profiler), has not yet been used very often, among other things because it has only recently emerged. 

Thus, for example, among the few examples found we see the work of Ascenso, Perkins and Williamon [25] with musicians from different countries (N = 601). They found a total mean of M = 7.34 (SD 1.68), highlighting among the five factors the vital purpose (M, meaning). That is, somewhat above the score obtained for the general population, according to the baseline measure carried out by Butler and Kern [28], who, for a sample of almost 32,000 people from different countries, found the following measures: P = 6.69 (SD 1.97); E = 7.25 (SD 1.77); R = 6.9 (SD 2.15); M = 7.06 (SD 2.17); A = 7.21 (SD 1.78). They obtained a global mean of M = 7.02 (SD 1.66). More recently, Ryan et al. [33], for their part, found in their PERMA-profiler validation study in the Australian adult population (N = 439), a total score of M = 6.6 (SD 1.5). 

Moreover, in the validation of the PERMA-profiler in Spanish, Pastrana and Salazar-Piñeros [34] found a total mean of the five factors of M = 7.73 (SD 1.37). The study was conducted with young men and women in Colombia who were volunteering (N = 230). Similarly, Lima-Castro et al. [35], for a sample of N = 1247 adults in Ecuador found a global mean of M = 8.17 (SD 3.12), somewhat superior to other studies, but with a very wide variance.

Furthermore, Cobo-Rendón et al. [36], found that among 1462 Chilean students with ages averaging 19, their PERMA values were certainly low, with an overall mean of M = 4.95 SD (1.4). While this may be surprising at first, it is consistent with many other studies related to age range and the happiness curve, mainly eudaimonic, which requires a certain maturity. Think, for example, of the vital purpose, for which it is necessary to have gone through a good part of life. It should be noted, on the other hand, that all these studies were conducted using the general PERMA-profiler scale, meaning that it was not adapted to the work environment. With this last version we only find the work of Watanabe et al. [31], who applied it to a heterogeneous sample of Japanese workers, obtaining a mean P = 5.46 (SD 2.3); E = 5.86 (SD 2.2); R = 5.59 (2.0); M = 6.24 (SD 2.1); A = 6.19 (1.9). We see how the results are notoriously inferior to those obtained in our study, although the difference may be due to the cultural and idiosyncratic distance of Japanese society (see Table 3). 

It should be noted that none of these studies under the PERMA model mentioned above showed statistically significant differences in relation to age. Nor did they appear in relation to gender, unlike our study, where differences in favor of women were found.

Some of the study’s limitations include sample size, for example. Although it is considerable and representative of the population studied, it is always desirable to achieve the largest possible sample size. On the other hand, it would have been interesting to complement the measurements with other scales, which would enrich the observation of the variables. Other types of socio-labor factors, such as income level, education, dependents, years in the company, etc., could also have been taken into account. Likewise, the measurement of other psychological factors, such as certain personality traits, values, ethical issues, etc., could also have contributed greatly. The truth is that, conversely, all these factors would have lengthened the questionnaire excessively, making it even more difficult to reach a reasonable sample, considering the time spent by the respondents on our questions, normally during their workday. Even so, we believe that this study mitigates, at least in part, the limited attention paid to the reality of the professionals and their day-to-day work, within the framework of ever-growing research on PR, as indicated, among others by Jiankun Guo and Anderson [37]. 

## 5. Conclusions

In short, one could say that public relations professionals, at least in a wide sample of Spanish practitioners, reasonably find all the necessary elements to achieve human flourishing in their profession, always following the PERMA model by Seligman [7]. They do so in a remarkable way, although perhaps it would be advisable to carry out a reflection as a sector, to consider how to achieve the margin of improvement indicated by the data. On the other hand, what some call the psychological capital of companies [38], which would characterize a truly positive organization [39], where a generalized leadership of an inspirational or transformational nature flourishes, which is one of the fundamental keys to organizational happiness [24,40,41,42]. 

Thus, in general terms, and carefully extrapolating the results obtained from the sample that represented the sector in this study, we can emphasize that:

Public relations professionals in Spain are reasonably happy in their work.

Women find greater happiness as PR professionals, especially because of greater engagement and relationships. However, age, along with the hierarchical level within the agency, do not affect the level of happiness. 

The main source of this happiness lies in the positive and enriching relationships that the professionals find in their environment, as well as the fact that they experience, in general terms, more positive emotions than negative ones during their day-to-day work. It should be noted that this is consistent with the fact that the social and working environment is one of the main sources of satisfaction [3] or dissatisfaction at the workplace.

Public relations professionals in Spain are happy in their work because they generally enjoy the tasks inherent in their work and their position, and they are committed to it in a remarkable way, an aspect that is related to another key issue: intrinsic motivation [1]. 

Public relations professionals in Spain are also happy in their work because their work offers the means of achieving professional goals and, therefore, personal self-fulfillment. 

Of the five factors that make up what we could call “professional flourishing”, in line with Seligman’s model [7], the vital purpose and its connection with the work is not at an unsatisfactory level, but it seems to be the main factor to improve. 

In relation to the last point, it should be stressed that if we seriously consider that we are at the beginning of an era in which having a real, sincere and honest purpose at the organizational level (beyond the purely monetary) is already a precondition, not only for the development of the organization but to a large extent for its survival. It should be noted that all this is being built at the personal and individual level, in the interest of that adjustment between the person and the organization, which is key to the flourishing of both [43]. That is to say, if an agency must help its clients to build a true corporate purpose, first it is necessary for the agency to have its own purpose clear, and its professionals too, and all of them aligned in a balanced way, as Bauer [44], or Rey and Malbašic [45], among many others, point out. When a person has a clear vital purpose as an individual and, to a great extent, that purpose is in accordance with his professional facet, the vital and labor fulfillment becomes more feasible. 

In regard to the main limitations of this paper, it is necessary to underline the sample size. A greater number of practitioners surveyed would have been desirable, but the difficult-to-collect responses from usually very busy practitioners were considerable. In addition to this, another limitation could refer to the fact that other causes of happiness are not studied, as well as neither other possible variables that could be influencing it, which could be an interesting field of future studies. On the other hand, as another way to develop future lines of research, it would be interesting to make the comparison between practitioners from different sectors, simultaneously and with the same tools. Only then it could be possible to infer differences in a statistically plausible way among job sectors. Finally, it is convenient to note that this research was developed just before the COVID-19 pandemic. It would be interesting to take this into account for comparative studies in the post-pandemic era. 

In any case, when asked whether public relations professionals are really happy in their jobs, we can answer that they are. However, we must also say that there is room for improvement. Therefore, the task ahead is to remember that people’s happiness is not only a key factor of professional and business success [46], but it is also an end in itself as stated by Biswas-Diener and Wiese [47]. In other words, what we can ultimately call the eudaimonic growth of a company involves the eudaimonic growth of all its stakeholders, starting with its professionals. It is here where the ethical dimension of this personal and organizational flowering is clearly evident, even though it has appeared merely in passing in the PERMA model used. Stortini [48], or Place [26], among many others, stress the need for ethics in a profession such as PR, the former reminding us of Kant’s words when he defined it as that science that teaches us not how to achieve happiness, but how to be worthy of it.

## Figures and Tables

**Table 1 ijerph-19-03987-t001:** Mean and SD obtained in the Workplace PERMA-profiler (range from 0 to 10), by gender, age and position.

			P	E	R	M	A	Total	N	H	Total with H and N
Gender	Male	Mean	6.4727	6.7212	7.3273	6.6970	7.1394	6.8715	4.9879	6.5636	5.1333
		SD	2.05950	1.87551	1.59667	1.89679	1.56832	1.59752	2.02247	1.95302	1.43708
	Female	Mean	7.1475	7.4754	8.2186	6.8169	7.5246	7.4366	4.7322	7.2295	5.6686
		SD	1.31534	1.28416	1.25366	1.65206	1.11169	1.03312	1.95667	1.59607	0.95133
Age	Under 32	Mean	6.9624	7.2151	8.4140	6.7097	7.5699	7.3742	4.4785	7.1828	5.6536
		SD	1.84766	1.51507	1.50007	1.62683	1.18489	1.23485	2.19174	1.90166	1.15322
	32–38	Mean	6.8182	7.1333	7.6909	6.6909	7.3152	7.1297	5.0061	7.0545	5.3853
		SD	1.43561	1.60195	1.30818	1.58030	1.18707	1.21617	1.83865	1.56184	1.08998
	Over 38	Mean	7.0222	7.3667	7.6833	6.9333	7.3167	7.2644	4.9778	6.8278	5.4532
		SD	1.51187	1.48907	1.34658	1.95986	1.44767	1.33045	1.84306	1.72113	1.19528
Position	High management	Mean	7.4348	7.7101	7.7826	7.3986	7.7246	7.6101	4.6667	7.0145	5.7712
		SD	1.33309	1.34552	1.32817	1.52933	1.21372	1.15990	1.56031	1.67768	1.04659
	Intermediate management	Mean	6.8611	7.3403	7.9306	6.6319	7.3542	7.2236	5.1042	6.9583	5.4246
		SD	1.40218	1.39653	1.19881	1.82087	1.20559	1.16493	1.93164	1.61058	1.09486
	Senior Technician	Mean	6.7821	7.0000	7.6026	6.6538	7.2308	7.0538	4.2564	7.5000	5.5018
		SD	1.61091	1.64924	1.58330	1.66662	1.26410	1.27164	2.00955	1.51217	1.14517
	Technician	Mean	6.7758	6.9636	8.3576	6.5636	7.3758	7.2073	4.9879	7.0061	5.4364
		SD	1.77088	1.44398	1.26174	1.60615	1.11121	1.12258	2.23603	1.81952	1.11095
Total		Mean	6.9379	7.2411	7.9416	6.7797	7.4049	7.2610	4.8117	7.0226	5.5023
		SD	1.60901	1.52823	1.42612	1.72724	1.27917	1.25921	1.97515	1.73700	1.14785

**Table 2 ijerph-19-03987-t002:** Significance coefficients of differences by gender, age and position.

	i	j	P	E	R	M	A	Total	N	H	Total with H and N	Happiness (Q16-i23)
Gender	Male	Female	0.079	0.029 *	0.000 *	0.895	0.236	0.069	0.507	0.040 *	0.027 *	0.610
Age	Under 32	32–38	0.880	0.955	0.015	0.998	0.547	0.553	0.320	0.918	0.422	0.805
		Over 38	0.991	0.848	0.011	0.903	0.371	0.803	0.380	0.388	0.510	0.939
	32–38	Over 38	0.817	0.692	0.999	0.884	0.963	0.906	0.988	0.650	0.983	0.952
Position	High management	Intermediate management	0.236	0.592	0.952	0.064	0.275	0.286	0.750	0.980	0.328	0.666
		Technician	0.144	0.049	0.129	0.069	0.483	0.318	0.844	1.000	0.426	0.331
		Senior technician	0.313	0.185	0.944	0.281	0.355	0.219	0.828	0.652	0.751	0.278
	Intermediate management	Technician	0.997	0.541	0.338	1.000	0.973	0.999	0.997	0.981	0.996	0.952
		Senior Technician	0.999	0.762	0.741	0.991	0.999	0.970	0.322	0.437	0.974	0.829
	Senior technician	Technician	1.000	0.541	0.077	0.996	0.959	0.947	0.398	0.617	0.994	0.973

* *p* < 0.05.

**Table 3 ijerph-19-03987-t003:** Key data of other similar studies.

Study	Sector	Country	N	M	SD
Ascenso, Perkins and Williamon [25]	Musicians	Several	601	7.34	1.68
Butler and Kern [28]	General Population	Several	32,000	7.02	1.66
Ryan et al. [33]	General Population	Australian	439	6.6	1.5
Pastrana and Salazar-Piñeros [34]	General Population	Colombia	230	7.73	1.37
Lima-Castro et al. [35]	General Population	Ecuador	1247	8.17	3.12
Cobo-Rendón et al. [36]	Students	Chile	1462	4.95	1.4
Watanabe et al. [31]	Several	Japan	310	5.88	1.8

## Data Availability

The data presented in this study are available on request from the corresponding author. The data are not publicly available due to confidentiality agreement.

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
