# Peer review of "Happiness at Work among Public Relations Practitioners in Spain"

_ijerph, 2022, doi:10.3390/ijerph19073987_

Round 1

Reviewer 1 Report

I read with great interest the paper "Happiness at work among public relations practitioners in Spain". The topic of the research is highly important, and the researched group is not that well studied. Yet I found several highly problematic issues, which inclined me to give a negative review. Below, I will shed more light on the subject.

  1. The theoretical introduction is very well prepared. The authors know the subject well, cite a broad selection of relevant papers, including scholars from different regions (that is, not limiting themselves to US and Spain only), which is a great advantage. On top of that, the quoted papers are new, indicating that the researchers are in line with the current development in the field. This is the strong side of the article.
  2. Main study assumption is not that clear. The title suggest a exploratory study, which is in strong contrast with the theoretical review. Either we do not know what is the state/base/relations of happiness at work with some concepts (due to lack of previous research or theory) and we conduct an exploratory study, or we know, but we want to verify the theory. The authors indicate the lack of knowledge (in the title), but then they provide sufficient background to put forward hypotheses.
    Let's assume that the title is a small error, and it is not an exploratory study. This way we can evaluate the hypotheses. The authors assume that there will be some differences in geneder (lines 105-106), and age (107-108) in the levels of happiness of PR workers. On top of that they suggest some mediating effects, but the maturity in the work would be key determinant of happiness. They do however reject/undermine those assumptions in the summary (line 118-119). There is still a different perspective put in lines 123-125, which suggest again that the confirmation of expectations towards job happiness requires time (amount not specified) spent in the profession. Those considerations, taken together are not clear. What is the single-minded goal of the study? What did the authors want to confirm? Verify? I can only imagine the introduction was changed to accommodate the lack of differences in the results, which is however a grave ethical error of HARK'ING (Hypotheses after results are known).
  3. Verfication of the hypotheses. Desipte unclear assumptions the authors go forward and try to verify their hypotheses. In my opinion, the method used to do that is wrong. I will briefly describe why:
    H1. PR is a profession that enables human flourishing...
    To verify this hypothesis, author would need to compare PR with other professions. Maybe all educated, urban professionals have similar levels of happiness? Maybe the payment level or type of employer permit such levels? We do not know, as there is no comparison group provided.
    H2, Female PR professionals are happier at work than their male counterparts
    The background to back this hypothesis is weak. Nevertheless the problem is with the results verification. The authors found significant differences between males and females on two (out of 5 in PERMA) dimensions. Is it sufficient? Did authors control for other factors (like types of relations, which are usually a stronger suit for females?) There is no trace of that.
    H3.The older the PR professional, the greater the happiness and human flourishing at work. - Main hypothesis, not confirmed in the study. The main question - were there enough older PR workers to even attempt a verification?
    H4: The position or hierarchical level affects the happiness of PR professionals at work. This could be a directional hypothesis (the higher the level, the greater the happiness...). Yet I am not sure, weathered there are enough levels (career steps) in the PR profession
    to be able to verify it. The authors fail to provide this crucial information in the introduction.
  4. To sum up - it looks like the authors had a chance to research pr professionals in their level of happiness, and would like to publish their results. Although this could be interesting, the research needs a serious reconsideration on the assumptions and hypotheses, before it can be published.

Author Response

Thank you very much for your very useful comment. Attached you can find the responses, apart from changes in the manuscript.

Reviewer 2 Report

First, let me say, I enjoyed this article.  It is logically constructed and readable.  I salute the authors’ command of English composition.

My few comments focus on editorial matters and I defer these matters to the editor and authors.  Nevertheless I shall mention a couple of these matters as I think they’ll improve the readability and impact of the manuscript.

  • Around lines 39-45 – I suggest these three points be bulleted. They are of sufficient importance to set them off for a reader.
  • Line 146 – capital “A” in “Appendix.”
  • The tables – Can the text be structured such that a table will not run across two pages? Table 2 is particularly annoying
  • 6, Table 2 – left side – the wrapping of text is annoying – this will need to be corrected for publication.
  • Around line 206 – a note that data were gathered pre-Covid is in order. The authors mention when data were gathered, earlier in the manuscript but reminding readers at this point would be useful.
  • Around line 226 – Is the n sufficient to support disaggregation by region, or by Madrid and outside of Madrid?
  • Maybe I missed it earlier but what was the response rate? To receive 256 responses, how many PR professionals were contacted?
  • Lines 269-296 – a table would make these data more accessible.

Author Response

(The authors gave the same response as above.)

Reviewer 3 Report

  1. The tables must be corrected because they are completely illegible in the presented form.
  2. It is worth examining the asymmetry of the grades distributions, because they are most likely tilted in the positive direction, which influences the inflated mean value, and then the presented conclusions regarding the optimistic assessment of happiness at work are not fully adequate. If this is the case (there is a right-hand asymmetry), then you can consider supplementing the information with the level of the median, which, as a position measure, is not sensitive to the presence of outliers.

Author Response

(The authors gave the same response as above.)

Reviewer 4 Report

Thank you for giving me an opportunity to review an interesting paper. I suggest some minor changes:

At the end of the introduction, add two paragraphs. In the first paragraph, explain what is the contribution of the paper, in relation to several previous papers - at least 1-2 references from scholarly journals, not later than 2010

In the second paragraph, describe other sections of the paper.

In the last section, please focus on “Conclusion” to include

(1).     Managerial and Academic Implications

(2).     Limitations of the paper

(3).     Future Studies and Recommendations

Author Response

(The authors gave the same response as above.)

Round 2

Reviewer 1 Report

Dear Authors,

Thanks for addressing some of my comments. I nevertheless still feel that the methodological part is underperforming.
My biggest concern, already put forward in the first review, concerns the hypothesis. The article presents a broad literature foundation, on which You assume, that the happiness of PR employees will depend on gender, age and tenure? Well, in my opinion, such hypotheses do not bring science forward. It is just ANOTHER group, where the same basic demographical variables would impact one dependent variable (happiness).
It is especially problematic, as You put into the study some variables You control (like health), which could be used as moderators/mediators of the posited effects. On top of that, the use of a 5-factor perma model should also be supported by specific hypotheses concerning the specific factors (else there is no point in using a five-factor measurement).
One last thing concerns the group - authors pose that the PR Profession is somehow specific (that is the reason they have selected it for research). What makes it specific? Where is this specificity addressed?

Bottom line - the hypothesis section needs a major rehaul. I specified the problems with the current hypotheses in the previous review. Now I only restate it that the text needs hypotheses touching on NOVEL areas of research, and using the broad theoretical background the authors provided.

Author Response

Dear Reviewer,

We do appreciate your comments and suggestions of reviewers. However, we disagree with several comments made in this last revision.

In our last revision we modified and addressed all suggestions made by the reviewers. We did 4 revisions and addressed all 4 reviews. However, in the second round, this revision leaves no room for improvement. Generally speaking, we disagree with it.

Your main concern are hypotheses. Our paper establishes some hypotheses to be contrasted, which we think is the scientific attitude. You say: “You assume, that the happiness of PR employees will depend on gender, age and tenure?” However, we do not make that assumption, we do not speak of “depend” but “relate”. Regarding the second hypothesis we stay that “Female PR professionals are happier at work than their male counterparts” and not that happiness depends on gender. We think these are two different things and we were very clear in our paper.

Regarding PERMA, it has indeed 5 factors, but it is a unitary model. You say that “the use of a 5-factor PERMA model should also be supported by specific hypotheses concerning the specific factors (else there is no point in using a five-factor measurement)”. We would agree with this point of view, but that doesn´t mean our approach is wrong. Your options are plausible but also ours using the PERMA model in its canonical unitary form.

Finally, we agree our results are modest but we are very confident that our work deserves honestly to be published and thus to continue de discussion with the community.

Sincerely,